Morpho-physiological effects of environmental stress on yield and quality of sweet corn varieties (Zea mays L.)

Tas Timucin 1 ttas_4@hotmail.com
Mutlu Arzu 2
1 Kepsut Vocational High School, Balikesir University , Balikesir, Marmara Region , Turkey
2 Akcakale Vocational High School, Harran University , Sanliurfa , Turkey
Capozzi Fiore
Electronic publication date: 2021 Dec 15
Publication date: 2021
Volume: 9
Electronic Location ID: e12613
Received 2021 Jun 8; Accepted 2021 Nov 18
Copyright: © 2021 Tas and Mutlu
Copyright year: 2021
Copyright holder: Tas and Mutlu
License: This is an open access article distributed under the terms of the Creative Commons Attribution License, which permits unrestricted use, distribution, reproduction and adaptation in any medium and for any purpose provided that it is properly attributed. For attribution, the original author(s), title, publication source (PeerJ) and either DOI or URL of the article must be cited.
License URL: https://creativecommons.org/licenses/by/4.0/

Keywords: Sweet corn, Heat stress, Pollen fertility, Abscisic acid

Funding: General Directorate of Agricultural Research Republic of Turkey Ministry of Agriculture and Forestry This study was funded under the national corn breeding program by General Directorate of Agricultural Research and Policies affiliated to Republic of Turkey Ministry of Agriculture and Forestry. The funders had no role in study design, data collection and analysis, decision to publish, or preparation of the manuscript.

==============================
Sweet corn is cultivated in different climatic regions of the world, and consumed either fresh or processed. Morpho-physiological effects of environmental stress on yield, yield components and quality of some sweet corn varieties were investigated in field experiments conducted at Southeastern Anatolia Region of Turkey during 2019 and 2020 growing seasons. The experimental lay out was randomized blocks with three replicates. Eight candidates and two control sweet corn varieties classified as moderate maturity (FAO 650–700) were used in field experiment. Mean values of pollen fertility rate (PFR, %), total soluble solids (TSS, °Brix), abscisic acid (ABA, nmol/g DW), ear length (EL, cm), plant height (PH, cm), number of grains per cob (CGN, grain) and fresh cob yield (FCY, t ha−1) were significantly different between years and sweet corn varieties. The PFR, TSS, ABA, EL, PH, CGN and FCY ranged from 40.29–67.65%, 13.24–20.09 °brix, 7.74–21.04 nmol/g DW, 9.69–15.98 cm, 97.80–171.34 cm, 289.15–420.33 grain and 4.15–10.23 t ha−1 respectively. The FCY, yield components and PFR values in the second year that had a higher temperature and lower relative humidity were lower compared to the first year, while ABA and TSS values were higher in the second year. Statistically significant correlations were recorded between the parameters investigated except FCY and TSS. The FCY and other parameters of sweet corn varieties, which produced high ABA phytohormone, were high, and the ABA hormone significantly contributed to plant growth under stress conditions. The results revealed that the PFR physiological parameter and ABA hormone in the plants provide important information about stress level and stress tolerance level of the cultivars, respectively. Despite adverse environmental stress conditions, the FCY of ŞADA-18.7 variety, one of the candidate varieties, was higher than that of the control and the mean value of the experiment.

Introduction

Sweet corn which is a type of maize harvested at the fresh stage and consumed as a vegetable. Sweet corn is belonging to family Poaceae and the genus Zea. Although sweet corn uses and accepts as a vegetable; maize, including sweet corn, is the third most important cereal crop after rice and wheat in the world. Smaller plant habitus and chemical composition of the grains are the distinguishing features of sweet corn from other corn types. The sugary1 (su1), shrunken2 (sh2), and sugary enhancer1 (se1 alleles) of sweet corn varieties prevent reduces the conversion of sucrose which is carried to the endosperm, to starch, and ensures that the grains have a high sugar content. The ratio of fat and protein is higher than other types of maize due to the larger relatively embryo of sweet corn (Szymanek, 2009).

Sweet corn is consumed before or after processing throughout the world. Unique taste, pleasant flavor and sweetness attracts the attentions of consumers. Sweet corn is also important for human nutrition due to positive contribution to health. In addition, high water (72.7%) and total solids (27.3%) contents of sweet corn kernels significantly contribute to the nutritional value. Furthermore, solid part contains hydrocarbons (81%), proteins (13%), lipids (3.5%) and others (2.5%) (Ozata, 2019). Starch is the dominant hydrocarbon component of solid part (Szymanek, 2012). Calorie content of sweet corn kernel is moderate compared to the other vegetables. The nutritional quality of sweet corn is also related to the dietary fiber, various vitamins, antioxidants, minerals, rich lutein, zeaxanthin and other carotenoids (Szymanek, 2012).

The sweet corn cultivation area in the world is about 1.6 million ha−1with approximately 20 million tons production and average yield of 13.029 t ha−1 (Anonymous, 2020). Contracted farming model is widely applied for the production of sweet corn especially in Aegean, Marmara and the western part of Turkey; however, domestic production doesnot meet the demand of the country. Therefore, frozen-canned sweet corn import of Turkey in 2018 was approximately 7,000 tons, which corresponds to approximately $9 million, while the export potential of sweet corn is only $0.85 million (Anonymous, 2018).

The adaptation of varieties to the ecological conditions of a region is important in selection of sweet corn varieties. Environmental factors which have adverse effects on plant growth and survival are known as abiotic stresses. The abiotic stress factors negatively affect the development of a plant, which adapted to any environment and grows smoothly (Abiko et al., 2005). High day and night time temperatures due to climate change threatens global agricultural production (Khaliq et al., 2019). Sweet corn is highly susceptible to environmental stress conditions compared to dent corn (Ordóñez et al., 2015). Several plant physiological and metabolic processes are adversely affected by heat stress, therefore heat stress is considered an important constraint to plant growth and crop productivity (Allakhverdiev et al., 2008). The responses of plants to abiotic stress factors vary depending on the amount of accumulation of phytohormones which modulate the physiological and molecular responses of plants to survive under abiotic stress conditions (Fahad et al., 2015). Absisic acid (ABA) is an important plant regulating hormone used to determine the tolerance of sweet corn plant to stress conditions (Islam et al., 2018), while, pollen vitality ratio also provides important clues about the stress levels of sweet corn plants. The ABA induces heat tolerance in corn (Islam et al., 2018). Similarly, Zandalinas et al. (2016) reported that ABA reduces the heat stress damage in sweet corn plants. Under heat and dry weather conditions, lack of ABA in corn plants increases water loss from stomata, therefore plants display heat stress-sensitive phenotypes (Du et al., 2013). Heat stress tolerance of mutants defective in ABA signaling (abi1-1) or lack of ABA biosynthesis (aba1-1) is reported very low (Suzuki et al., 2016).

The optimum temperature for maize flowering occurrs between 25 °C and 35 °C (Sánchez, Rasmussen & Porter, 2014). Furthermore, the optimum temperature for maximum maize grain yield is around 25 °C. Although the temperature tolerance level of maize varieties varies among themselves, it has been reported that maize varieties experience high stress and struggle to survive at temperatures above 40 °C (Alam et al., 2017). Although the effect of temperature stress on number of grains in pre-tasseling period is not clear, high temperature in tasseling-silking interval leads to significant yield reduction (Lizasoa et al., 2018). Environmental factors such as fertility of soils, available water content, soil and air temperatures and relative humidity have significant effects on pollens production ability of plants; therefore, they impact the size and number of seeds produced. Pollens have 60% water content by weight; and, are sensitive to environmental conditions (Firon, Nepi & Pacini, 2012). Environmental stress affects plant growth and even more drastically male reproductive development and causes to form sterile pollens (Begcy et al., 2019). The exposure of heat stress during micro gametogenesis cause abortion of microspores and male sterility (Rieu, Twell & Firon, 2017; Begcy & Dresselhaus, 2018). Alghabari et al. (2014) indicated that increased temperatures during the microspore stage of sweet corn pollens reduced corn cob fertility by affecting endogenous levels of ABA. Exposure of 35 °C light and 25 °C dark stress for 48 h specifically at pollination stage negatively affects gas exchange and chlorophyll content in addition to pollen fertility (Sunoj et al., 2017). Heat stress and low relative moisture humidity reduce pollen viability; thus they can be used to distinguish crop genotypes against various infield stresses (Razzaq et al., 2019). The seed yield is affected by the extreme temperatures. Temperatures greater than or equal to 32 °C are reported decreasing the viability of pollens, retention of pollens in the anthers and germination of pollens in various sweet corn varieties (Razzaq et al., 2017). High temperature during silking, pollination and grain filling profoundly affects pollen shedding and viability of sweet corn pollens. Therefore, pollen viability and filled grains or number of grains under stress conditions have a positive correlation (Kalyar et al., 2013). The sugar content in kernels of sweet corn determines the quality of sweet corn (Szymanek, Tanaś & Kassar, 2015). Varieties, climate and harvest time have significant effects on sugar content of kernels (Abadi & Sugiharto, 2019). Eating quality of final product is directly related to tenderness of sweet corn genotypes with a higher sugar and lower starch contents and an increased aroma (Ibrahim & Ghada, 2019). The sugar content of fresh kernels was reported higher compared to the frozen and canned grains (Alan et al., 2014).

Although sugar corn is a summer-growing plant and a total temperature requirement is between 10.000 °C and 12.000 °C during a growing season, sugar corn is stressed at temperatures above 35 °C, resulting in losses in terms of yield, some agronomic and quality values (Tiwari & Yadav, 2019). Sweet corn production which is mainly carried out in the western regions of Turkey, should be spread to the rest of the country to decrease sweet corn import and economic losses. Southern Turkey meets about 17% of dent corn production and is dominated by the high temperatures and dry weather conditions. Selection and breeding of sugar corn varieties that may successfully be grown in the southwest regions and tolerate high temperatures are highly important to expand the sugar corn production in Turkey. In this study, sweet corn varieties tolerant to environmental stress conditions were determined, and the relationship between physiological and biochemical responses of sugar corn varieties with fresh cob yield, some agronomic and quality characteristics were investigated.

Material & Methods

The study was conducted between June and November during the second crop corn season in 2019 and 2020. The experimental field was located in South Eastern Project Agricultural Research Institute, and lied between 36°44′10″ N latitudes and 38°50′20″ E longitudes (Fig. 1).

Figure 1 Geographical position of the study area.

The location having the hottest and driest climatic conditions on the Syrian border of Turkey was selected in both years of the experiment. Map data: Google, © 2021 CNES/Airbus, Maxar Technologies.

Soil and climate characteristics of the study area

Soils of the experimental site have clayey texture, pH varies from 7.96 to 8.00 and electrical conductivity is between 1.05 and 1.40 dS/m which indicates no salinity problem Lime content is between 29.2% and 32.3% and increases with soil depth. Plant available phosphorus and organic matter contents were between 10.7–28.8 kg/ha−1 and 0.73–1.24%, respectively.

The study area is located in a semi-arid climate zone with high temperatures and low relative humidity in summers. The temperature and relative humidity in the second year were significantly higher compared to the first year. The highest temperature recorded in June, July, August and September for both 2019 and 2020 was over 40 °C and relative humidity was less than 40%. In the first year of the study, total precipitation during growing season was 27.8 mm/m-2, while it was only 0.4 mm/m-2 in the second year. Long term climate data indicated severe abiotic stress conditions such as high temperature and low relative humidity in the region. Current climate data revealed that, the amount of precipitation decreased and the semi-arid nature of the region has evolved into an arid region (Table 1).

Table 1 Important climatic values of the trial location.

Months	Mean temperatures (°C)	The highest temperatures (°C)	The lowest temperatures (°C)	Mean humidity (%)	Total precipitation (mm/m−2)	
	2019	2020	ALY	2019	2020	ALY	2019	2020	ALY	2019	2020	ALY	2019	2020	ALY	
May	31.0	32.2	28.7	38.8	40.0	40.3	15.8	14.6	12.5	42.3	45.9	51.2	10.2	0.2	26.8	
June	36.5	37.9	34.6	41.6	44.2	44.1	18.9	21.4	15.3	34.4	33.9	36.2	0.8	0.0	4.3	
July	38.9	41.6	38.8	42.2	44.7	46.8	21.9	24.1	18.0	35.7	31.1	33.7	0.2	0.0	2.0	
August	39.7	40.1	38.4	42.8	45.4	46.2	22.7	22.0	19.0	38.9	36.2	32.5	1.2	0.0	3.4	
September	35.5	38.3	34.0	38.6	43.4	43.9	17.7	19.7	13.0	40.0	42.3	29.9	0.0	0.2	4.6	
October	30.7	32.2	27.1	34.6	37.0	37.8	12.9	14.9	10.9	50.6	39.5	43.1	12.8	0.0	26.5	
November	22.6	24.1	18.7	26.5	28.1	30.8	6.9	10.9	1.1	47.2	43.3	52.8	2.6	0.0	45.1	
Average	33.6	35.2	31.5	37.9	40.4	41.4	16.7	18.2	12.8	41.3	38.9	39.9	27.8	0.4	112.7	
Note:

ALY, Average for long years (1920–2020).

The average temperature during the 12-day top tasseling period in both years was over 40 °C. Daily average temperature values in both years are shown in Fig. 2.

Figure 2 Mean daily temperature values during the tasseling period in both years of the study.

The climate data used in the experiment were obtained daily from the climate station in the area where the experiment was conducted.

Experimental design and varieties traits

Eight candidates and two control sweet corn varieties classified as moderate maturity (FAO 650–700) were used in the study. The origins, grain colors and hybrid crosses of the varieties are shown in Table 2. The experiment was started at the second week of June in 2019 and 2020. Experimental lay out was randomized blocks with three replicates. Sweet corn seeds were planted in 70 cm interrow and 18 cm intrarow spacing. Each plot had four rows with 5 m length and total plot area was 14 m−2. Each plot had 110–115 plants.

Table 2 Sweet corn varieties and their characteristics.

Variety name	Sweet corn	Ear colour	Origin	
ŞADA-18.1	super sweet (su x sh)	yellow	Turkey	
ŞADA-18.2	super sweet (su x sh)	yellow	Turkey	
ŞADA-18.3	super sweet (su x sh)	yellow	Turkey	
ŞADA-18.4	super sweet (su x sh)	yellow	Turkey	
ŞADA-18.5	super sweet (su x sh)	yellow	Turkey	
ŞADA-18.6	super sweet (su x sh)	yellow	Turkey	
ŞADA-18.7	super sweet (su x sh)	yellow	Turkey	
ŞADA-18.8	super sweet (su x sh)	yellow	Turkey	
Control-1	super sweet (su x sh)	yellow	USA	
Control-2	super sweet (su x sh)	yellow	USA	

Fertilization

Fertilizer was applied in the seedbeds at rate of 400 kg ha−1 in the form of diammonium phosphate (18%N-46%P2O5-0%K20) during planting and the rest of the nitrogen (400 kg N ha−1) was applied in the form of urea (46% N into two equal parts) at the second and thirteenth weeks of planting (Seydosoglu & Cengiz, 2020).

Irrigation

Sprinkler irrigation method was used after planting to ensure a homogeneous seedgermination. Furrow irrigation method was applied in subsequent irrigations. The amount and number of irrigation during vegetation period varied depending on climate and water demand of plants. In the first year of the study, 600 mm of water was given in five irrigations, while in the second year 670 mm of water was given in six irrigations.

Cobs harvest

Fresh cobs were harvested between 17 and 25 September in both years of the trial. Harvest time of varieties varied between 80 and 90 days. Fresh cobs were harvested when the grain moisture content was between 70% and 75% (Ozata, 2019).

Analytical procedures

Ten fresh cobs were collected from experiental plots, and the husks were peeled to determine the number of grains per cob (CGN, number), ear length (EL, cm), and fresh ear yield (FEY, t ha−1) (Ibrahim & Ghada, 2019).

Total soluble solids (TSS, °Brix)

Sweet corn kernels were squeezed out, and total soluble solids (TSS) content in sweet corn kernel juice was determined using a hand refractrometer (0–32 °Brix). The TSS value of each cob was recorded, and the TSS value representing the sweet corn variety was determined by calculating the mean values (Alan et al., 2014).

Pollen fertility rate (PFR, %)

Three tassels of each sweet corn variety were collected from each plot between 10:00 p.m. and 14:00 p.m. during tasseling development period when the anthers begin to dehisce. The samples of tassels were quickly transported to the laboratory in an ice. The pollens grains brought to the laboratory were placed on a glass lamella using a fine-tipped brush. A drop of 1% TTC (2.3.5 Triphenyl Tetrazolium Chloride) liquid, which contained 0.2 g TTC and 12 g sucrose dissolved in 20 mL distilled water, was dropped on pollen samples, and the pollens were dyed within 2 to 3 h. Pollen grains were counted under a light microscope to determine the viability rate of the dyed pollens. Pollen grains not dyed with TTC (dark red or brown color) were considered not viable, while pollen grains dyed with orange or bright red color were evaluated as viable (Sulusoglu & Cavusoglu, 2014).

Absisic acid (ABA, nmol/g DW)

In the post-tassel emergence period, 10 flag leaves which were the last leaves to emerge in a cereal plant were sampled randomly in each plot. One hundred mg leaf sample was weighed and wrapped with aluminum foil and transported to the laboratory at −198 °C in a nitrogen tank. Leaf samples were stored in a refrigerator at −20 °C. The ABA level of leaf samples was determined using Enzyme-Linked Immuno Sorbent Assay (ELISA) method (Cabot, Poschenrieder & Barcelo, 1986).

I) Preparation of the extraction buffer

To prepare the extraction buffer; 400 ml of methanol (99%) and 0.85 sodium bicarbonate (NaHCO3) were dissolved in deionized water and the final volume was adjusted to 100 ml. Five mg of butylated hydroxytoluene was added to a 100 mM NaHCO3 solution. Then, 400 ml of methanol (99%), 5 mg of BTH and 100 mM NaHCO3 were mixed together and the final volume was adjusted to 500 ml. The pH of the extraction buffer was adjusted to 8.0 using HCl and stored at 40 °C.

II) Plant extraction stages

Leaf samples in liquid nitrogen was ground in a mortar and 3 ml of extraction buffer (1.5 + 1.5 ml) was added and homogenized. Homogenization was carried out in a dark environment. All samples were transferred to falcon tubes without leaving any herbal extract residue in the mortar. The homogenate was transferred to 15 ml falcon tubes and incubated at 4 °C in the dark for 48 h.

Homogenates in falcon tubes (3 ml) were equally distributed with the pellets after 48 h in two separate sterile 2 ml ependorfs. All ependorfs were centrifuged at 4 °C for 30 min. Supernatants (upper liquid part) in ependorf tubes were transferred to 15 ml clean falcon tubes and the 1st supernatant was obtained. One ml of cold extraction buffer was added to the pellets remaining in the eppendorf. Following the vortexing, the solution was incubated in the dark for 24 h at 4 °C. Afterwards, incubated solution was centrifuged at 4 °C for 30 min and the second supernatant was obtained. The 2nd supernatant was added to the falcon tubes containing the first supernatants and the final volumes in the falcon tubes were recorded in a table.

Heat block instrument was placed in a fume hood and ventilation was run. The temperature of the heat block was set to 45 °C. Each time, 12 falcon tubes were placed in the heat block and the liquid part was evaporated. The falcon tubes, whose liquid part was partially or completely evaporated, were closed and stored at −20 °C until the ELISA test.

III) ABA reading stages

The chemicals and materials in the test kit consisted of standard (64 nmol/ml), standard dilution solution, ELISA micro plate (96 well), Str HRP conjugate reagent, 30X wash solution, biotin ABA, choromogen solution A, choromogen solution B, stop solution, plate closure membrane and sealed bags. The ABA solution (64 nmol/ml) in the ELISA KIT was diluted in 0, 2, 4, 8, 16 and 32 nmol/ml and standards were prepared. The 0 nmol/ml standard was set as a blank. The 50 μL of Streptavidin-HRP was pipetted into the standard wells of 96-well micro plate, and 40 μL of sample + 10 μL of ABA antibody + 50 μL of streptavidin-HRP was pipetted into the wells containing samples prepared by plant extraction.

After all wells were completed (empty wells were marked), the micro plate was covered with a membrane, gently shaken and incubated for 60 min at 37 °C. Meanwhile, 2.5 ml of the washing solution was taken and diluted with 72.5 ml of distilled water. The membrane covered on the micro plate was gently lifted and the liquids inside the plate drained 350 μL of the previously prepared washing solution was added to each well, and after 1–2 min, the plate in this solution was turned down and removed.

A total of 50 μL of choromogen A and 50 μL of choromogen B were pipetted into all wells, including blanks, and gently shaken. The solutions were incubated for 10 min at 37 °C in the dark. Then, 50 μL of stop solution was added in all wells to stop the reaction. This step was performed when blue color in the wells turned into yellow. Auto zero was performed with the solution in the blank wells and the micro plate was placed into the reader. ABA values were obtained by reading the absorbance values at 450 nm with the Enzyme-Linked Immuno Sorbent Assay (ELISA KITS) method (Fig. 3).

Figure 3 The ABA level of leaf samples was determined using Enzyme-Linked Immuno Sorbent Assay (ELISA)method.

Stages such as homogenization process of 100 mg leaf tissue (A) and transfer of homogenates to 15 ml falcon tubes (B) and the falcon tubes with mixtures of 1st and 2nd supernatants (C) and microplate which is consisting of 96-well for ABA readings (D) of ABA analysis in sugar corn leaves.

Statistical analysis

Statistical analysis was performed using JUMP (Version 13.2.0) statistical software. Analysis of variance including degrees of freedom and mean squares of all parameters in combined years was performed. The mean values for the varieties were compared using the LSD test at P ≤ 0.05 probability level. Correlation test was performed between all traits determined in the study (Ucak et al., 2016).

Results

The difference between years was statistically significant (P ≤ 0.01) for PFR, ABA and significant (P ≤ 0.05) for PH, EL, CGN, TSS. The difference between varieties was statistically significant (P ≤ 0.01) for all paremeters. The difference for PH, EL, CGN, FCY and PFR paremeters in year x variety interaction was not significant, while it was significant (P ≤ 0.05) for TSS and ABA (P ≤ 0.01). Mean squares and degrees of freedom analysis of variance for yield and some traits of sweet corn varieties in combined years are shown in Table 3.

Table 3 Mean squares and degrees of freedom analysis of variance results on yield and some traits of sweet corn varieties in combined years.

Source	DF	Mean squares	
PH	EL	CGN	FCY	PFR	TSS	ABA	
Year	1	5,363.82150*	220.30168*	19,797.67020*	112.17603*	455.84241**	193.64474*	185.22294**	
Repetition (year)	2	116.89560	5.77886	1,048.98562	6.47530	9.28743	6.61809	1.49086	
Variety	9	3,585.55343**	30.57739**	12,063.18190**	22.69901**	531.35760**	39.37639**	103.27515**	
Year x Variety	9	56.51974 ns	2.37874 ns	680.67828 ns	0.65450 ns	12.35434 ns	6.65239*	8.80407**	
Error	38	80.47829	1.33520	504.59526	1.18801	7.24173	3.14695	2.87081	
CV		6.78	9.00	6.58	15.17	5.07	10.35	11.03	
Mean		132.24	12.82	341.21	7.18	53.04	17.13	15.37	
Notes:

* Significant at 0.05 level of probability.

** Significant at 0.01 level of probability.

ns, not significant; DF, degrees of freedom; CV, coefficient of variation.

Pollen fertility ratio (PFR), total soluble solids (TSS) and absisic acid (ABA) levels of sweet corn varieties

The PFR and ABA parameters were significantly different (P ≤ 0.01) between years and cultivars. The difference in TSS parameter was very significant (P ≤ 0.01) between sweet corn varieties, while it was significant (P ≤ 0.05) between the years (Table 4).

Table 4 Pollen fertility and total soluble solids and absisic acid of sweet corn varieties in 2019 and 2020.

The performance of sweet corn varieties in terms of ABA, TSS and PFR.

Varieties	Pollen fertility rate (%)	Total soluble solids (°brix)	Absisic acid (nmol/g DW)	
2019	2020	Mean	2019	2020	Mean	2019	2020	Mean	
ŞADA-18.1	44.33 f	38.48 h	41.41 gh	17.32 abc	18.51 cd	17.92 bc	8.04 f	12.11 ef	10.08 g	
ŞADA-18.2	45.51 ef	35.07 h	40.29 h	17.70 ab	20.15 bc	18.93 ab	5.67 f	9.81 f	7.74 h	
ŞADA-18.3	58.14 bc	55.22 cd	56.68 d	12.63 f	15.24 de	13.94 e	15.44 bcd	18.40 bc	16.92 cd	
ŞADA-18.4	46.22 ef	42.37 g	44.29 g	16.29 bcd	19.66 bc	17.98 bc	11.67 e	14.26 de	12.96 f	
ŞADA-18.5	53.89cd	50.66 e	52.28 e	14.59 cdef	19.88 bc	17.24 bc	14.77 bcd	17.15 bcd	15.96 de	
ŞADA-18.6	50.41 de	46.70 f	48.56 f	14.41 def	18.33 cd	16.37 cd	12.79 de	15.78 cd	14.28 ef	
ŞADA-18.7	61.44 b	58.55 bc	60.00 c	15.70 bcde	24.47 a	20.09 a	14.33 cde	24.11a	19.22 ab	
ŞADA-18.8	57.36 bc	52.93de	55.15 de	13.37 ef	16.15 de	14.76 de	19.42 a	22.67 a	21.04 a	
Control-1	68.85 a	59.37 b	64.11 b	11.91 f	14.56 e	13.24 e	17.14 ab	19.59 b	18.37 bc	
Control-2	71.81 a	63.48 a	67.65a	19.45 a	22.34ab	20.90 a	16.82 abc	17.34 bc	17.08 cd	
Mean**	55.80 a	50.28 b	53.04	15.34 b	18.93 a	17.13	13.61 b	17.12 a	15.37	
CV (%)	5.34	4.17	5.07	10.90	10.34	10.35	11.67	10.41	11.03	
LSD (0.05)	5.11**	3.15**	2.87**	3.36**	2.07**	2.72**	3.06**	1.98**	
Mean LSD	3.39**	2.86*	1.36**	
Notes:

The means indicated with the same letter in the same column are not significantly different according to the JUMP test at P ≤ 0.05.

The means indicated with the same letter in the same row are not significantly different at P ≤ 0.05.

The means of different year-varieties combinations with the same lower case letters are not significantly different according to the JUMP test at P ≤ 0.05.

* Significant at 0.05 level of probability.

** Significant at 0.01 level of probability.

The PFR parameter is one of the most important abiotic stress indicators for sweet corn varieties. The highest mean PFR value was obtained from Control-2 (67.65%) sweet corn variety, while the lowest PFR value was obtained from ŞADA-18.2 (40.29%) variety. The mean PFR in the first year was 55.80%, while the mean PFR in the second year was 50.28%. High temperature and low relative humidity values in the study area caused approximately 30–50% decrease in PFR values of sweet corn varieties. Daily average temperatures above 40 °C especially during 12-day of tasseling period negatively affected the pollen fertility. The results of correlation test indicated that the increase in PFR caused an increase in ABA level and yield components such as FCY, PH, EL and CGN.

The highest TSS value (20.09 °brix) was recorded from the SADA-18.7 variety, and the lowest TSS value (13.24 °brix) was obtained from the Control-1 variety. The Obrix values of most sweet corn varieties were higher than the mean value of the experiment. High TSS values in sweet corn varieties are desired, but not yield losses. The TSS parameter had a negative correlation with PFR, FCY and yield components, while the TSS had a positive correlation with ABA.

The ABA level recorded in ŞADA-18.8 sweet corn candidate variety (21.04 nmol/g DW) was higher than control and other varieties. Sweet corn varieties produced ABA hormone in both years of the study. The differences in environmental conditions (temperature and relative humidity) between years have also caused the differences in ABA levels. The results revealed that the amount of ABA hormone produced by sweet corn varieties varies depending on environmental stress conditions and genetic characteristics of the varieties. Positive correlation between ABA hormone and all other parameters indicates that the increase in ABA hormone causes an increase in FCY, yield components, PFR and TSS values of the sweet corn varieties.

Some agronomic traits of sweet corn varieties

The ear length (EL) and plant height (PH) values in 2019 and 2020 were statistically significant at P ≤ 0.01 level, while the difference between years was significant only at P ≤ 0.05 level (Table 5).

Table 5 Agronomic traits performance of sweet corn varieties in 2019 and 2020.

The performance of sweet corn varieties in terms of EL and PH.

Varieties	Ear length (cm)	Plant height (cm)	
2019	2020	Mean	2019	2020	Mean	
ŞADA-18.1	12.62 ef	6.76 e	9.69 e	107.90 h	93.33 gh	100.62 h	
ŞADA-18.2	12.15 f	9.20 d	10.68 de	111.90 gh	83.70 h	97.80 h	
ŞADA-18.3	15.21 bc	11.87 bc	13.54 b	152.79 cd	138.30 bc	145.54 cd	
ŞADA-18.4	13.36 def	7.98 de	10.67 de	121.90 fg	103.63 fg	112.77 g	
ŞADA-18.5	14.22 cd	10.09 cd	12.15 c	129.57 ef	114.48 ef	122.02 fg	
ŞADA-18.6	13.77 cde	8.65 de	11.21 cd	136.90 e	120.78 de	128.84 ef	
ŞADA-18.7	17.69 a	14.28 a	15.98 a	168.56 b	148.52 ab	158.54 b	
ŞADA-18.8	14.68 cd	12.72 ab	13.70 b	141.90 de	129.41 cd	135.65 de	
Control-1	17.16 a	13.79 ab	15.48 a	158.56 bc	139.85 bc	149.21 bc	
Control-2	16.55 ab	13.76 ab	15.16 a	186.90 a	155.78 a	171.34 a	
Mean**	14.74 a	10.91 b	12.82	141.69 a	122.78 b	132.24	
CV (%)	5.96	12.47	9.00	5.53	5.99	6.78	
LSD (0.05)	1.51**	2.33**	1.35**	13.45**	12.61**	10.49**	
Mean LSD	2.67*	12.01*	
Notes:

The means indicated with the same letter in the same column are not significantly different according to the JUMP test at P ≤ 0.05.

The means indicated with the same letter in the same row are not significantly different at P ≤ 0.05.

The means of different year-varieties combinations with the same lower case letters are not significantly different according to the JUMP test at P ≤ 0.05.

* Significant at 0.05 level of probability.

** Significant at 0.01 level of probability.

The EL and PH parameters are important agronomic traits which may have direct or indirect relationship with fresh cob yield (FCY). The highest EL value was measured in ŞADA-18.7 (14.28 cm) candidate variety and Control-1 and control-2 varieties were placed in the same statistical group with ŞADA-18.7. The lowest EL values was recorded in ŞADA-18.1 (6.76 cm). Lower EL values in the second year can be attributed to the higher temperature and lower relative humidity values.

The mean PH value of sweet corn varieties in the second year was 122.78 cm, while it was 141.69 cm in the first year. Similar to the EL parameter, PH values were also affected by differences in temperature and relative humidity between years. The highest PH value was recorded in Control-2 cultivar (171.34 cm), while the lowest PH value was measured in ŞADA-18.2 cultivar (97.80 cm).

Number of grains per cob and fresh cob yields of sweet corn varieties

The differences in number of grains per cob (CGN) and FCY in both years of the experiment were statistically significant at P ≤ 0.01 level, and the differences between years were significant at P ≤ 0.05 level. Statistical groups of varieties and years in terms of CGN and FCY parameters and level of significance were given in Table 6.

Table 6 A number of grains/per cob and fresh cob yield of sweet corn varieties in 2019 and 2020.

Important parameters such as FCY and CGN.

Varieties	A number of grains/per cob (grain)	Fresh cob yield (t ha−1)	
2019	2020	Mean	2019	2020	Mean	
ŞADA-18.1	299.07 g	295.00 bcd	297.04 e	6.64 ef	4.13 ef	5.38 fg	
ŞADA-18.2	312.52 fg	265.78 d	289.15 e	5.49 f	2.82 f	4.15 g	
ŞADA-18.3	367.07 bc	314.78 bcd	340.93 bc	9.40 bc	6.86 c	8.13 cd	
ŞADA-18.4	328.25 ef	322.63 b	325.44 cd	7.39 def	5.20 de	6.29 ef	
ŞADA-18.5	354.22 cd	301.29 bcd	327.76 cd	8.26 cde	4.92 de	6.59 ef	
ŞADA-18.6	339.44 de	271.23 cd	305.33 de	7.66 cde	3.46 f	5.56 f	
ŞADA-18.7	428.37 a	412.29 a	420.33 a	11.34 a	9.11 a	10.23 a	
ŞADA-18.8	353.59 cd	317.30 bc	335.44 bc	8.63 bcd	5.99 cd	7.31 de	
Control-1	378.11 b	337.41 b	357.76 b	10.21 ab	7.13 bc	8.67 bc	
Control-2	433.15 a	392.78 a	412.97 a	10.49 ab	8.55 ab	9.52 ab	
Mean**	359.38 a	323.05 b	341.21	8.55 a	5.82 b	7.18	
CV (%)	3.17	8.85	6.58	12.95	14.55	15.17	
LSD (0.05)	19.53**	49.07**	26.25**	1.90**	1.45**	1.27**	
Mean LSD	35.98*	2.83*	
Notes:

The means indicated with the same letter in the same column are not significantly different according to the JUMP test at P ≤ 0.05.

The means indicated with the same letter in the same row are not significantly different at P ≤ 0.05.

The means of different year-varieties combinations with the same lower case letters are not significantly different according to the JUMP test at P ≤ 0.05.

* Significant at 0.05 level of probability.

** Significant at 0.01 level of probability.

The CGN parameter is an important yield component and is directly related to the yield. Similar to the other yield components, the CGN parameter was negatively affected by the extreme temperature and relative humidity values. The mean CGN value of sweet corn varieties in 2020 was 323.05 grains, when environmental stress conditions were dominant, the CGN was 359.38 grains in 2019. The highest CGN value during the experiment was obtained from ŞADA-18.7 (420.33 grains), while the lowest CGN value was recorded in ŞADA-18.2 (289.15 grains) sweet corn variety.

The results of yield components were in agreement with the results of FCY parameter. The highest FCY value was obtained in ŞADA-18.7 (10.23 t ha−1) candidate variety, and the lowest FCY value was recorded in ŞADA-18.2 (4.15 t ha−1) variety. The FCY values recorded in ŞADA-18.8, ŞADA-18.3 and ŞADA-18.7 candidate varieties were higher than the mean values of the experiment. The FCY value obtained in ŞADA-18.7 variety was higher than the control varieties.

Some parameters measured in the study were significantly different between years. Daily average temperatures during the tasseling period were above 40 °C, which decreased the pollen viability ratios by almost half. Low relative humidity ratios in the same period also had an impact on the decrease in pollen viability. The ABA and TSS values in 2020 were higher, while the FCY values were lower, due to higher temperature and lower relative humidity compared to the first year of the experiment. The mean values of PFR, ABA, TSS and FCY in both years were shown in Fig. 4.

Figure 4 The mean values in both years of Some parameters such as polen fertility rate (PFR, %), Absisic asit (ABA, nmol/g DW), Total soluble solids (TSS, °brix) and Fresh cob yield (FCY, t ha−1) (P ≤ 0.05).

The sweet corn varieties investigated in the study showed different physiological and biochemical responses under stress conditions such as severe temperature and low relative humidity.

The images of normal and abnormal pollens were recorded during pollen examination under the light microscope. The light red and bright colored pollens in the image A are alive, while the mat and black pollens in the image B are not alive. Pollen images of ŞADA-18.7 sweet corn variety examined under light microscope are given in Fig. 5.

Figure 5 Pollen fertility (A) and pollen infertility (B) images of ŞADA-18.7 from sweet corn varieties under light microscope.

Pollen grains were counted under a light microscope to determine the viability rate of the dyed pollens. Pollen grains not dyed with TTC (dark red or brown color) were considered not viable, while pollen grains dyed with orange or bright red color were evaluated as viable.

Correlations

All parameters of sweet corn varieties determined were subjected to correlation test. The results of correlation test including correlation coefficients and level of significance was given in Table 7. Significant positive correlations were recorded between EL and PH (r = 0.8162, P < 0.01), CGN and PH (r = 0.8292, P < 0.01), CGN and EL (r = 0.7154, P < 0.01), FCY and PH (r = 0.8577, P < 0.01), FCY and EL (r = 0.8666, P < 0.01), FCY and CGN (r = 0.8434, P < 0.01), PFR and PH (r = 0.9128, P < 0.01), PFR and EL (r = 0.7664, P < 0.01), PFR and CGN (r = 0.7796, P < 0.01), PFR and FCY (r = 0.7691, P < 0.01).

Table 7 Correlation coefficients and significance levels of yield components and grain yield and physiological parametre.

Traits	Traits	Correlation coefficients (r)	Count	The lowest coefficients (95%)	The highest coefficients (95%)	Significance levels (P)	Correlation levels	
EL	PH	0.8162	60	0.7092	0.8864	<0.0001**		
CGN	PH	0.8292	60	0.7287	0.8947	<0.0001**		
CGN	EL	0.7154	60	0.5640	0.8203	<0.0001**		
FCY	PH	0.8577	60	0.7718	0.9128	<0.0001**		
FCY	EL	0.8666	60	0.7855	0.9184	<0.0001**		
FCY	CGN	0.8434	60	0.7501	0.9037	<0.0001**		
PFR	PH	0.9128	60	0.8576	0.9472	<0.0001**		
PFR	EL	0.7664	60	0.6363	0.8541	<0.0001**		
PFR	CGN	0.7796	60	0.6555	0.8628	<0.0001**		
PFR	FCY	0.7691	60	0.6402	0.8559	<0.0001**		
TSS	PH	−0.1769	60	−0.4123	0.0806	0.1763		
TSS	EL	−0.3503	60	−0.5549	−0.1058	0.0061**		
TSS	CGN	−0.0167	60	−0.2695	0.2382	0.8992		
TSS	FCY	−0.2775	60	−0.4964	−0.0253	0.0319*		
TSS	PFR	−0.1891	60	−0.4227	0.0681	0.1479		
ABA	PH	0.4701	60	0.2455	0.6468	0.0002**		
ABA	EL	0.2579	60	0.0043	0.4804	0.0466*		
ABA	CGN	0.3627	60	0.1199	0.5647	0.0044**		
ABA	FCY	0.2888	60	0.0377	0.5057	0.0252*		
ABA	PFR	0.5285	60	0.3171	0.6898	<0.0001**		
ABA	TSS	0.0519	60	−0.2047	0.3019	0.6936		
Notes:

*,** Significant at 0.05 and 0.01 levels of probability respectively.

FCY, Fresh Cob Yield (t ha−1); CGN, Number of Grains/Per Cob (grain); PH, Plant Height (cm); EL, Ear Length (cm); PFR, Pollen Fertility Rate (%); TSS, Total Soluble Solids (°brix); ABA, Absisic Acid (nmol/g−1 DW).

Discussion

Agronomic traits such as PH and EL are important parameters that provide information on FCY. Taller sweet corn plants have a higher number of leaves. Plants with higher number of leaves cary out more photosynthesis, which causes an increase in EL and FCY values. Similar to the FCY and CGN, the PH and EL values decreased in the second year when environmental stress conditions such as high heat and low relative humidity were more dominant. It is thought that the plant’s nutrient and biomass accumulation were insufficient under heat stress, due to disruptions in vital plant activities such as stomal conductivity, photosynthesis efficiency, chlorophyll content in leaves and insufficient gas exchange. It could be said that there were decreases in parameters such as PH, CGN, EL and indirectly FCR, due to these problems in plants. Stress conditions such as high temperature and low relative humidity experienced during the trial years suppressed the growth of plants, as a result of this, vegetative stages of some plants were completed very fast, and thus healthy leaves could not be developed and healthy plant height did not occur. The characteristics of a variety and climate have significant impact on plant heights of sweet corn (Subaedah, Edy & Mariana, 2021).

The highest correlation coefficients of the experiment were obtained between PH and FCY, CGN and EL. The sweet corn varieties with the tallest plants had high FCY, EL and CGN values. Positive correlation between PH and PFR and ABA indicated that sweet corn varieties with a good PH had high PFR and ABA values. The plant heights recorded in this study are in harmony with the plant heights (130.83–171.67 cm) of Subaedah, Edy & Mariana (2021), while they were lower than the plant heights (170.8–197.8 cm) reported by Ibrahim & Ghada (2019).

Similar to the PH, EL parameter is a directly related yield component with FCY, PFR and CGN. The results indicated that EL values of the varieties with high pollen viability rates were also high, therefore the FCY value was also higher. The EL parameter had a negative correlation with TSS, while the EL had significantly positive correlations with other traits. The FCY values of sweet corn varieties, which have a long cob size were also high. The EL values recorded are compatible with the findings of Kara (2011), while they are lower than the EL values reported by Ibrahim & Ghada (2019) (16.2–21.4 cm). Genetic characteristics of the varieties and environmental stress conditions of the study played a determinant role in the EL values measured. Short cob sizes can be attributed to the decrease in photosynthes under stress conditions such as temperature and low relative humidity and insufficient production of nutrients in the leaves.

Positive correlation between CGN and PFR parameters in sweet corn varieties indicated that the CGN is an important parameter to assess the impacts of environmental factors on sweet corn. The results showed that the fertilization rates of sweet corn varieties were high when viable pollen numbers in the grains were high, and in that case, the number of grains formed on a cob were higher. The CGN and EL, PH, FCY and PFR parameters were highly correlated and a moderate positive correlation was recorded between CGN and ABA. The results revealed that PFR values, which are important physiological parameters, coincided with the CGN values of sweet corn varieties. Due to the high temperatures of over 40 °C during the tassel flowering period of the sugar corn plants, the PFR values decreased between 30% and 50% in both trial years. In addition, it is assumed that the synchrony between the tasseling and silking of plants is disrupted. Consistent with our study, high heat stress at the reproduction phase negatively affects the physiology of plants like flower initiation, source-sink relationship, falling of pods, which ultimately decreases the number of grains (Cairns et al., 2013). Similar to our findings, Lizasoa et al. (2018) reported that the number of grains per cob in sweet corn varieties had a high correlation with yield and yield components. Harmony in our findings, Patel & Mankad (2014) reported a significant correlation between high cob yield and pollen fertility of sweet corn varieties. Consistent with our findings, Lizasoa et al. (2018) observed a 42% decrease in grain weight per plant and 32% in pollen viability with the increases in day/night temperatures from 25/15 °C to 35/15 °C during tasseling-silking interval period.

Although the FCR values changed between years, heat stress negatively affected the fresh cob yields in both years. The PFR values obtained in both years give us clues about the fresh cob yields. Furthermore, tissue injuries in leaves are predicted to adversely affect the rate of photosynthes during heat stress, which can cause leaf damage and increase leaf senescence that largely result in decreasing photosynthetic efficiency. It is assumed that decreasing photosynthetic rate also reduced other yield components, especially FCR.

TSS parameter is one of the determining quality parameters of sugar corn stress conditions, especially high temperature, contributed positively to TSS values. Contrary to the yield and yield components, the TSS values in the grains of sweet corn varieties were higher in the second year when warmer and dry weather conditions prevailed compared to the first year of the experiment. Due to high temperature stress in the second year of the trial, plants cannot absorb the water lost through evapotranspiration through the roots, Insufficient water content at cellular level in plants caused an increase in the dry matter ratio, in other words TSS values. Environmental stress causes decrease in yield and yield components, while increase in TSS values. The changes in TSS values of sweet corn varieties may attributed to the genetic structure and environmental conditions such as temperature, sowing time and harvest time.

Similar to our findings, Ibrahim & Ghada (2019) reported that TSS contents of sweet corn hybrids ranged from 12.10 to 17.43 °Brix. Alan et al. (2014) recorded higher TSS values (16.3 to 27.4 °Brix) in kernels of seven sweet corn varieties. The results of Subaedah, Edy & Mariana (2021), which reported that the TSS values of sweet corn varieties varied between 20.8–22.8 °Brix under Indonesian conditions, are substantially similar to those obtained in this study. The researchers associated the changes in TSS values to the differences in climate and variety.

ABA is a stress hormone that accumulates in leaves and roots, depending on the stress conditions and variety. The positive correlation between ABA and yield components indicates that the ABA hormone produced by plants promoted both lateral shoot and lateral root development; therefore, plants with strong roots and shoots grown healthier. The findings of Opitz et al. (2016) and Seeve et al. (2017) who reported that severe environmental stress inhibited both root and shoot growth but ABA promoted the formation of lateral roots and shoot, are in accordance with our conclusion. The results indicated that ABA hormone is an important selection criterion of sweet corn varieties or lines that are tolerant to environmental stress conditions. ABA plays a vital role in plants’ physiological adjustments such as against heat stresses along with increasing seedling growth, endogenous levels of ABA and reduced oxidative damage to plants due to heat stress (Meena & Yadav, 2015). Similar to our findings, the importance of ABA as a phytohormone involved in heat stress response and tolerance has been reported by Wasilewska et al. (2008) and Haizhen et al. (2018). In addition, Xu et al. (2013) indicated that ABA accumulates under stress conditions. In line with our study, several researchers had suggested that the ABA is an important direct heat-tolerant selection criterion in crops (Iqbal et al., 2017). Tao & Zhao (2010) indicated that Plant growth hormone such as ABA play important roles in strengthening heat tolerance in maize under high-temperature stress. ABA induces maize to produce heat shock proteins (HSPs), strengthening photosystem II (PSII) heat tolerance (Maestri, Klueva & Perrotta, 2002). In another study conducted, an exogenous ABA increases the maize cell membrane antioxidant capacity to improve heat tolerance (Gong et al., 1997). Phytohormone such as ABA, have key roles in coordinating various signal transduction pathways in plant during the heat stress response has been reported by Wani et al. (2016). Hasanuzzaman, Gill & Fujita (2018) observed that ABA is a signaling molecule and also enhance the number of other signaling molecules such as nitric oxide for thermos-tolerance.

The values of physiological parameters such as PFR and ABA determined in this study are compatible with the FCY values. Significant positive correlations were obtained between FCY and yield components such as PH, EL and CGN and physiological parameter such as PFR. Moderate positive correlation was recorded between FCY and ABA, while a negative correlation was obtained between FCY and TSS. The researchers showed that yield and yield components were decreased under high temperature conditions. Sweet corn varieties grown in the experimental area where environmental stress conditions prevail presented a good performance in terms of FCY. The responses of sweet corn varieties to environmental stress conditions and genetic characteristics of the varieties can be related to the good performance.

Values of ŞADA-18.7 candidate sweet corn variety were better compared to the other varieties, and the highest yield of the experiment was obtained from ŞADA-18.7 candidate variety. The differences in the yield of the tested sweet corn varieties was related to the differences in genetic potentials. Subaedah, Edy & Mariana (2021) also indicated that the variety and the climate conditions of the exponential field have significant impact on growth and yield of sweet corn varieties. The high yield of sweet corn varieties was supported by the number of grains per cob, as well as the ear length. Therefore, higher number of grains per cob and a longer cob size cause to obtain higher yield per unit area (Khan, Khan & Afzal, 2017). Ilker (2011) also indicated that the fresh cob yield value was positively correlated with the yield components. The findings are consistent with the results obtained by Ahmad et al. (2015) who showed that fresh ear yields of sweet corn had a significant positive correlation with CGN, PH, EL and yield components.

Conclusions

The results of the study indicated that pollen fertility is an important index of environmental stress. The PFR parameter has come to the fore as an important and practically usable selection criterion to determine the most suitable varieties under stress conditions. The ABA hormone has a significant role in regulating the tolerance level of plants to temperature stress conditions. The ABA hormone protects the plants against heat stress at the cellular level, in addition, the ABA hormone positively contributes to the yield components, especially FCY and PFR values. The ABA hormone level will provide important clues about the tolerance level and yield of the sweet corn varieties in breeding programs to be carried out under abiotic or biotic stress conditions in the future, or in adaptation studies to determine the appropriate variety for all regions in the world. In long-term breeding and adaptation programs, the ABA hormone level of sweet corn lines and varieties included in breeding programs should be determined to save time and labor. The results concluded that physiological parameters such as ABA and PFR will save time and labor in breeding programs and variety adaptation experiments carried out with multiple materials in areas with environmental stress such as high temperature and low relative humidity.

Supplemental Information

Supplemental Information 1 Raw data.

Click here for additional data file.

Additional Information and Declarations

Competing Interests

Author Contributions

Data Availability

The authors declare that they have no competing interests.

Timucin Tas conceived and designed the experiments, performed the experiments, analyzed the data, prepared figures and/or tables, conducted statistical analysis and field studies, and approved the final draft.

Arzu Mutlu conceived and designed the experiments, performed the experiments, analyzed the data, authored or reviewed drafts of the paper, made chemical analysis in the laboratory, and approved the final draft.

The following information was supplied regarding data availability:

The raw measurements are available in the Supplemental File.

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
