# Peer review of "Morpho-physiological effects of environmental stress on yield and quality of sweet corn varieties (Zea mays L.)"

_PeerJ, doi:10.7717/peerj.12613_

## Round 0.1 · original submission · Major Revisions

The manuscript could be accepted after the authors have addressed the major issues raised by the three reviewers. In particular (not exclusive), add more details in the "Methods section", be more specific in highlighting the study problem and make the hypothesis clearer. In addition, Improve the results and discussion section as also suggested by REV#2.

·

Basic reporting

English is generally good, but does need editorial work. I made numerous suggestions, but it will need more work.

Literature and structure are appropriate. Raw data is shared.

results are relevant to hypothesis, although the hypothesis is not clearly stated. This can be handled in the rewrite stage.

Experimental design

It appears to be original research. I leave it to the editors to decide whether the research was within the Aims and Scope of the journal. I really hope that they had decided that before they ask me to spend my time reviewing the paper.

Research questions are well defined and the need for the research is made clear.

My only significant concerns are within this box.

1. Are two years (environments) adequate for this type of work. I leave that question to the editors. I do believe that the results reflect what would be expected in those two seasons.

2. The authors must supply more information regarding the germplasm used in this experiment and also more detail regarding the statistical analysis. Without the knowledge about the germplasm it is impossible to determine th generalizability of the data and repeat the experiment.

Validity of the findings

I believe the authors have done well in addressing this area,

Reviewer 2 ·

Basic reporting

The language of reporting is not clear and lacks scientific integrity.

Experimental design

Satisfactory

Validity of the findings

Methods section needs more details, so that the reproducibility of results could be ensured.

Additional comments

There should not be repetition of title words in abstract or any other section. Abstract is poorly written, please use standard way to write abstract section. No objectives and methods mentioned in abstract.

The introduction section is too lengthy and so much general information. Please be specific and highlight the study problem, how present study will fill the gap and answer which questions. What are the specific and general objectives.

Why such a high dose of N (400 kg) was applied to plots? What was the exact does of NPK?

L203 when the 10 cobs were sampled?

L219 write the exact date of sampling.

Give more details about the statistical procedures.

If the authors had written Results and discussion as separate sections, then no need to justify results in results section, please remove the cited references.

In the correlation section, please just mention the strong relations and the ones which are of interest, no need to mention the values as already mentioned the tables/figures.

The discussion section is too brief and general. Authors are advised to please elaborate the underlying mechanisms and reasoning while justifying their results.

L394, please clearly mention the range of high temperature which is lethal to crop plants.

Figure 1 gives no or poor information about site.

Reviewer 3 ·

Basic reporting

In this manuscript, the authors studied the effects two environmental stresses, temperature and humidity, on the morpho-agronomic traits of several sweet corn varieties doing the experiments in two successive years, 2019 1nd 2020. The work is interesting and data are promising. However, there are some major issues, especially in writing, as I pointed out below:
1. Although the authors have provided the key results of this study in the abstract, analysis and overall discussion are not sufficient to understand what are the stress conditions considered and how they affected the results. Since the title of this paper is “effects of stress on …..”, the content of the abstract should be improved accordingly to get a better reflection of the title in the results.
2. L#83-84: “High water (72.7%) and total solids (27.3%) contents of sweet corn kernels significantly contribute to the nutritional value”. Do the authors indicate both moisture and solid as “high”? How can both be high simultaneously in the same kernels?
3. The unit “nmol/g-1 DW” should be either “nmol/g DW” or nmol g-1 DW”
4. Please make proper sub-sectioning in “materials and method” sections. Provide the information on the varieties used in this study and more details on the design of experimentation (plot size, plant numbers in each row, replications etc.). How the data were compared statistically among the varieties and between the years?
5. Results section: analysis and presentation of the results should be stress-oriented rather than emphasizing individual varieties or years.

Experimental design

The authors mostly considered temperature and humidity as the environmental stresses in this study. However, they did not control these conditions artificially as it was a field study and fully depended on the natural fluctuations in two seasons. More importantly, as per the data presented in Table 1, such fluctuations seem to be not so significant. Based on the overall context and aim of this work, my major concern is, why did the authors selected the same time period (June-November) of two cropping years if they really wanted to investigate the effects environmental stress on the morpho-agronomy of the plants? Wouldn’t it be better to consider different seasons for studying such effects?

Validity of the findings

Good

Additional comments

As above

---

## Round 0.2 · accepted · Accept

Many thanks to the authors for their excellent work in having considered and integrated the comments of the reviewers in this last version of the manuscript.. Congratulations

Reviewer 2 ·

Basic reporting

Good.

Experimental design

Good

Validity of the findings

Good

Additional comments

Now the authors have revised their manuscript with all the suggested changes, it can be considered for publication.

Reviewer 3 ·

Basic reporting

The authors have made substantial efforts in addressing the reviewer's comments and revising the manuscript. Therefore, I recommend acceptance of the revised manuscript.

Experimental design

Good

Validity of the findings

Okay

Additional comments

None